# Management of Transcatheter Aortic Valve Implantation and Complex Aorta Anatomy: The Importance of Pre-Procedural Planning

**DOI:** 10.3390/ijerph19084763

**Published:** 2022-04-14

**Authors:** Alfredo Intorcia, Vittorio Ambrosini, Michele Capasso, Riccardo Granata, Fabio Magliulo, Giannignazio Luigi Carbone, Stefano Capobianco, Francesco Rotondi, Francesca Lanni, Fiore Manganelli, Emilio Di Lorenzo

**Affiliations:** Department of Cardiology and Cardiovascular Surgery, “San Giuseppe Moscati” Hospital, 83100 Avellino, Italy; vitt.ambrosini@gmail.com (V.A.); capasmi@libero.it (M.C.); riccardogranata@hotmail.com (R.G.); fabmagl87@hotmail.it (F.M.); giannignaziocarbone@gmail.com (G.L.C.); capobianco.stefano@gmail.com (S.C.); francesco.rotondi@tin.it (F.R.); franclanni@hotmail.it (F.L.); fioremang@gmail.com (F.M.); emidilorenzomd@gmail.com (E.D.L.)

**Keywords:** transcatheter aortic valve implantation, S-shaped Aorta, aorta tortuosity, buddy wire technique

## Abstract

Aortic stenosis is the most common primary valve lesion requiring surgery or, especially for older patients, transcatheter intervention (TAVI). We showcase a successful transfemoral TAVI procedure in a very high-risk patient and an extremely tortuous S-shaped descending aorta, characterized by heavy calcifications and multiple strong resistance points. We demonstrated that transfemoral TAVI using the “buddy stiff guidewire” technique could be a feasible, simple, quick, and easy procedure able to straighten an extremely abdominal aorta tortuosity. With all techniques available and careful pre-procedural planning, and thanks to the flexibility of new generation TAVI delivery systems, it is possible to safely perform the procedure even in the most challenging patients.

## 1. Introduction

Aortic stenosis (AS) is the most common primary valve lesion in both Europe and North America, requiring surgical or transcatheter intervention, and is rapidly increasing as a result of the aging population [1]. It is a degenerative disease with a prevalence of >7% among patients aged >80 years [2]. Symptomatic severe aortic stenosis has a poor prognosis; in patients who develop symptoms such as exertional dyspnoea, angina, or syncope, the two-year mortality rate is as high as 60% [3]. In fact, there are no medical therapies that have been shown to delay the progression of AS, and aortic valve replacement (either surgical or percutaneous) remains the only treatment that has been shown to improve survival. Therefore, both American and European valve heart guidelines strongly recommend aortic valve replacement (AVR) and early intervention in all patients with a high transaortic echocardiographic gradient (a mean gradient of ≥40 mm Hg or a peak velocity of ≥4 m/s, class I recommendation), regardless of the left ventricular ejection fraction (LVEF). The best modality of intervention for a patient undergoing AVR for symptomatic aortic stenosis over the years has become increasingly complex, thanks to the continuous technological advances such as minimal access surgery with rapid deployment valves, or the latest generation devices for Transcatheter Aortic Valve Implantation (TAVI). For this reason, the current guideline recommends evaluating all the potential cases of AS in a Heart Valve team as part of a Heart Valve Centre, that carefully consider the decision regarding the indication, timing, and modality of AVR between a surgical aortic valve replacement (SAVR) approach versus TAVI. Since the first human case of a transcatheter aortic valve replacement in 2002 [4], the role of TAVI has expanded from treating inoperable and/or prohibitive high-risk patients suffering from severe symptomatic AS, through intermediate-risk cases, and now, based on the latest clinical trial results, even low-risk cases. TAVI has clearly become the preferred therapy over SAVR in a setting of high-risk patients with symptomatic and severe AS, a mean Society of Thoracic Surgeon (STS) score of >8% and has been shown to be non-inferior (or even superior when transfemoral access is feasible) to surgical AVR in a clinical setting of intermediate-risk patients with an STS score between 4 and 8% [5]. In addition, results from the latest two randomized trials comparing TAVR with SAVR in low-risk patients (STS score of <4%) have reported that TAVI may also be superior to surgical AVR in the short period of a one-year follow up, although longer-term data on the evaluation of outcomes, especially regarding the valve durability of TAVI, are still lacking [6,7]. In the TAVI procedure, age plays a key role; according to the latest European Society of Cardiology (ESC) guidelines [1], TAVI is mainly recommended for all elderly patients (≥75 years) at higher surgical risk (STS/EuroSCORE II of >8%), or for patients who are not suitable for surgery. In our case, we want to demonstrate that a successful transfemoral TAVI can be performed in a very high-risk patient considered unsuitable for surgery and with multiple severe comorbidities such as active lung cancer and an extremely complex S-shaped anatomy of the descending aorta. We show that TAVI can be performed even in the most difficult patients [8] initially considered contraindicated for aortic valve replacement, thanks to thoughtful planning, careful procedures, and all currently available techniques.

## 2. Case Presentation

We analyze the case of a 75-year-old male patient who presented to our Cardiovascular Department with chest pain and dyspnoea (New York Heart Association functional class III). He had a recent diagnosis of early stage IIA non-small-cell (NSCLC) lung cancer (considered suitable for chemotherapy and surgery by our oncologist with a 5-year survival rate of 60%). He had a long history of diabetes, hypertension, and also suffered from extreme spinal deformity due to severe scoliosis. The transthoracic echocardiogram showed preserved left ventricular function and severe aortic stenosis with a mean pressure gradient of 48 mm Hg, a valve area of 0.7 cm^2^, and a peak aortic jet velocity of 4.3 m/s. In addition to this, the pre-procedural multislice computed tomography (CT) showed extreme tortuosity with an anatomy of the calcified descending aorta with a double S-type curve, adding a more complex scenario to any potential AVR treatment. The extreme tortuosity of the descending aorta is clearly visible in the CT-images both in a coronal view, with an acute angle of 75°, and in the sagittal view, with an acute angle calculated of 95° (Figure 1A–D). As recommended by our oncologist team, an aortic valve replacement is a mandatory procedure in the special case of a patient who has to start chemotherapy treatment for lung cancer. Therefore, he was referred to the Heart Team, where TAVI was considered the most suitable procedure compared to surgical aortic valve replacement (SAVR), in view of the extreme risks of cardiac surgery (calculated STS score was 8.4%, Euroscore II was 24.59%). Through a detailed analysis of multimodal imaging and in particular the CT, it was possible to assess the best way to obtain access (femoral in this case) and to verify the anatomical relationships between the aortic valve and the coronary root or ostia. It was also crucial to choose the optimal device size (we considered in this case a 23 mm SAPIEN 3 Transcatheter Heart Valve-THV for a native annulus diameter of 20 mm and a native annulus area of 400 mm^2^) or, as in this case, to plan the optimal technique for crossing the S-shaped descending aorta before the procedure. Given the anatomical tortuosity, we thought that the use of a double stiff guidewire, the so-called “buddy wire” technique, could straighten the descending aorta and reduce any resistance points. On the day of the TAVI, an echo-guided bilateral femoral arterial access (9 Fr on the right side and 7 Fr on the left side) was obtained, together with a 7 Fr left femoral venous access for transvenous pacing. We first performed a standard coronary angiography examination that showed nearly normal coronary arteries. Subsequently, through the left femoral arterial sheath, a 0.035 inch Back-up Meier^TM^ steerable guidewire (Boston Scientific, Marlborough, MA, USA) was inserted into the proximal descending aorta (Figure 2A) to start stretching the vessel. From the right side, we first placed a Supra Core 35 Hi-Torque 0.035. Extra Support guidewire (Abbott Vascular, 3200 Lakeside Drive, Santa Clara, CA, USA) and, over this guidewire, switched the 9 Fr introducer with a 14 F Edwards eSheath (Edwards LifeSciences, Irvine, CA, USA). Through this sheath, a 0.035 inch INNOWI SX^®^ preformed guidewire (SYMEDRIX GmbH, Oberhaching, Germany) was inserted into the left ventricular chamber. At this point, with the two extra-rigid guidewires within the lumen of the aorta, able to straighten the tortuosity of the vessel, a 23 mm SAPIEN 3 THV was carefully advanced (Figure 2B) and finally deployed under high-speed burst pacing. The Back-up Meier “support” guidewire was held in place until the delivery system was able to pass through the tortuosity points of the abdominal aorta and then quickly removed to avoid some complications, such as aortic dissections or perforations. At the end of the entire procedure, an aortogram demonstrated a good valve position and no evidence of paravalvular leakage or complications (Figure 2C). The patient was immediately taken to the Intensive Care Unit and discharged from the Cardiovascular Department after just three days in NYHA Class I.

## 3. Conclusions

There are rare cases [8,9] in the medical literature showing the performance of TAVI with techniques to straighten the vessel in patients with extreme tortuosity of the thoraco-abdominal aorta, such as an “S” conformation, as well as in those at very high surgical risk due to severe comorbidities, such as active lung cancer. As already explained, the decision involving the indication and modality of intervention between SAVR and TAVI procedures deserves careful pre-procedural planning in a very high surgical risk patient such as this. The assessment of operative risk has been facilitated through the use of conventional scoring systems, useful to evaluate the risks of cardiac surgery, such as the Society for Thoracic Surgeons Predicted Risk of Mortality (STS-PROM) and the European System for Cardiac Operative Risk Evaluation (EuroSCORE II) risk calculator systems. However, although these scores are accurate in identifying patients at the highest risk, they were developed to predict the average mortality rate within 30 days after cardiac surgery and have been found to overestimate in-hospital mortality after the TAVI procedure. For this reason, the current guidelines for the management of aortic valve disease recommend considering other risk factors beyond the conventional scores mentioned above in the decision making of patients undergoing TAVI. In this scenario, the assessment of “frailty”, defined as a “syndrome of impaired physiological reserve and increased vulnerability to stressors” [10,11], becomes an important predictor of poor outcomes post-TAVI procedure. In the management of complex patients suffering from aortic valve disease, this is very important, as already stated by the recent ESC consensus document, the presence of a multidisciplinary Heart Valve Team as a part of a Heart Valve center hospital. In such context, the indications of the guidelines suggest choosing AVR in patients at low-risk with an STS-PROM or EuroSCORE II score of <4% or EuroSCORE I logistic of <10% and no other risk or technical factors such as frailty, porcelain aorta, or chest radiation sequelae. Patients with associated cardiac conditions requiring concomitant surgery, like severe complex coronary artery disease, ascending aortic aneurysm, severe primary mitral or tricuspid valve disease, septal hypertrophy, or the presence of aortic or left ventricular thrombi, should also be evaluated for a surgical AVR. Some others anatomical features, like the presence of an unsuitable aortic root anatomy with low coronary heights (defined as a distance from the coronary ostium to the aortic valve anulus of <12 mm), or an extreme annular dilatation, or a dysmorphic valve morphology with a presence of a bicuspid aortic valve leaflets with various degree and pattern of calcification, should all be considered from the Heart team suitable for SAVR. On the other hand, according to the latest guideline, TAVI is recommended for patients judged unsuitable for SAVR by the Heart Valve Team (Class I of the ESC guideline), particularly patients at higher surgical risk (STS-PROM or EuroSCORE II score of ≥ 8% or EuroSCORE I logistic of ≥10%), and especially elderly patients (>75 years) with suitable access for transfemoral TAVI. Despite the possibility of alternative peripheral accesses (transapical, trans-subclavian, etc.,) we considered, like all the current guidelines, the femoral access as the gold standard for TAVI and we recommend reconsidering surgery when this access is not feasible [12]. For this purpose, we chose a balloon-expandable Sapien3 THV prosthesis, that is associated with lower rates of pacemaker implantation after the procedure, and its Ultra-Low profile 14 Fr Edwards eSheath (Edwards LifeSciences, Irvine, CA, USA) introducer system. This sheath expands transiently during the passage of the Sapien 3 THV facilitating the placement of the large femoral sheath into the atheroma-altered arterial wall of the iliaco–femoral axis and the extremely tortuous descending aorta of the patient. The assessment of the Heart Team, consisting of a cardiac surgeon, an interventional cardiologist, a clinical cardiologist, and cardiac imaging is therefore crucial in this complex scenario not only to choose the best treatment option for the patient but also for careful pre-procedural planning. In this case, TAVI was the preferred procedure over SAVR because the combination of STS/EUROSCORE of >8 points, severe aortic wall calcifications, an age of >75 years, and the frailty of the patient with coexisting active lung cancer were all features in favor of transcatheter aortic valve implantation [13,14]. Vessel anatomy that was thought to be prohibitive even just a few years ago can be overcome thanks to improvements in delivery system technology, better procedural planning, and a greater operator experience. Of course, caution must always be taken in these extreme cases because tortuosity can predispose to rupture or arterial wall dissection, which can be a dramatic event when the aorta is involved. In particular, the presence of calcifications (some calcified atherosclerotic plaques are clearly visible in Figure 1B,D in the 3D-rendering CT view along the descending aorta and both the iliac and femoral arteries) may increase the risk of rupture or dissection due to the reduced compliance of the aortic wall. Fortunately, in the literature, the reported incidence of acute aortic dissection during TAVR is a rare complication involving about 0.2% of cases [15]. We routinely perform an aortography of the ascending aorta at the end of the procedure to avoid the above complications. In a case like this, it is very important to choose the best treatment option for the patient to give them a chance of surviving lung cancer. The extremely tortuous descending aorta was characterized by several points of strong resistance, which led to great difficulty in pushing and delivering the valve device. For this reason, and thanks to an accurate analysis of the pre-procedural angio–CT images, we have previously considered in the Heart Team performing TAVI with the help of the so called “Buddy Wire” technique, an approach derived from the experience of treating severe calcifications in percutaneous coronary interventions (PCI). The Buddy Wire technique consists of a second guidewire (usually a 0.014 inch guide wire) placed next to the one used to advance balloons and stents into the coronary artery during the percutaneous coronary intervention (PCI). The “Buddy Wire” is a simple, fast, and readily available procedure. In this case, the use of a second 0.035 inch “stiff” guidewire (instead of a 0.014 inch coronary guidewire), alongside the first wire used to advance the valve device during TAVI, should be chosen to straighten an extremely abdominal aorta tortuosity. Obviously, this technique is useful only in the case of severe complex aorta anatomy, while the presence of a second stiff guide wire lying in the aorta without necessity can lead to some complications such as aortic dissection or perforation. Transfemoral TAVI with the “Buddy double-stiff guidewire” (as we have named it) technique can be feasible only if correctly used [16,17,18,19]. With this approach, we demonstrate that TAVI can be performed substantially safer and can reduce the likelihood of procedural complications even in high-risk patients with a very complex anatomy. With all existing techniques and thanks to the flexibility of the new generation TAVI delivery systems, it is possible to safely perform this procedure even in the most difficult patients, who were initially considered contraindicated to aortic valve replacement.

## Figures and Tables

**Figure 1 ijerph-19-04763-f001:**
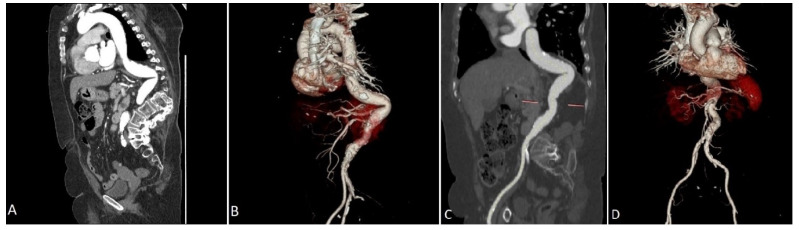
Preprocedural Computed Tomography reconstruction of the arterial tree demonstrating an extreme tortuous descending aorta and spinal cord deformation in a coronal view (**A**) and in a 3D rendering volume view (**B**). Is clearly visible the maximum acute angle of the tortuosity calculated in 75° and the relationship with calcium distribution. (**C**): a sagittal view through the aorta demonstrated a calculated maximum angulation of 96°. (**D**): a 3D rendering volume reconstruction clearly shows the abdominal aorta and surrounded anatomical structures.

**Figure 2 ijerph-19-04763-f002:**
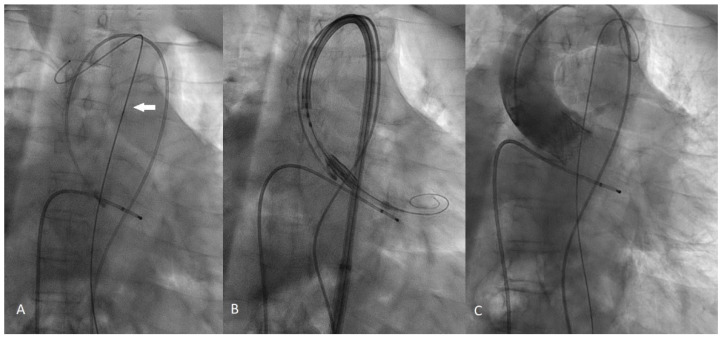
Intraprocedural fluoroscopy steps. (**A**): The Back-up Mayer (Boston Scientific, USA) 0.035 inch stiff guidewire (white arrow) placed along the aortic arch helps to straightens the aorta. (**B**): The Sapien 3 Heart valve prior to deployment; it is clearly visible that the INNOWY SX (SYMEDRIX GmbH) second stiff guidewire is positioned in the left ventricular chamber. (**C**): The aortography clearly demonstrated the correct deployment of the Sapien 3 valve without any evidence of paravalvular leak.

## Data Availability

Data regarding this clinical case can be provided by the authors following reasonable request.

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
