# Peer review of "Management of Transcatheter Aortic Valve Implantation and Complex Aorta Anatomy: The Importance of Pre-Procedural Planning"

_ijerph, 2022, doi:10.3390/ijerph19084763_

Round 1

Reviewer 1 Report

I felt that TAVR was performed very well in a case with very strong aortic tortuosity.

I would like to make a few minor comments.

  • This is a case of lung cancer. How long is the prognosis of life expected with treatment(chemotherapy and surgery)?
  • Have you considered alternative access other than femoral access?
  • Please add the annulus size and the selected Sapien3 size in this case.
  • Please comment on the difference between cases in which straightening of aorta can be expected using the buddy wire technique and those in which it cannot be expected, based on the CT scan performed prior to the procedure.

Author Response

1. This is a case of lung cancer. How long is the prognosis of life expected with treatment (chemotherapy and surgery)?

He had a recent diagnosis of early stage IIA non-small-cell (NSCLC) lung cancer, considered suitable for chemotherapy and surgery by our oncologist with a 5-year survival rate of 60%.

2. Have you considered alternative access other than femoral access?

Despite the possibility of alternative peripheral accesses (transapical, trans-subclavian, etc.) we considered, like all the current guidelines, the femoral access as the gold standard for TAVI and we recommend to reconsider surgery when this access is not feasible.

3. Please add the annulus size and the selected Sapien3 size in this case.

We considered a 23 mm SAPIEN 3 THV prosthesis for a native annulus size of 20 mm and a native annulus area of 400 mm2

4. Please comment on the difference between cases in which straightening of aorta can be expected using the buddy wire technique and those in which it cannot be expected, based on the CT scan performed prior to the procedure

The pre-operative angio-CT images allowed us to plan the best way to treat this patient and in particular to choose the use of a second stiff guidewire lying in Aorta (the so called "Buddy wire technique") to solve the problems of the severe multiple Aorta kinking. Obviously, this technique is useful also in the case of severe complex Aorta anatomy, while the presence of a second stiff guide wire lying in Aorta without necessity can lead to some complications suh as aortic dissection or perforation.

Reviewer 2 Report

Intorcia et al. reported a successful transfemoral TAVI procedure in a highly tortuous S-shaped descending Aorta using a buddy stiff guidewire” technique. The authors mentioned that the buddy stiff guidewire” technique could be feasible, simple, quick, and easy procedure able to straighten a significantly abdominal Aorta tortuosity.

My comments are below.

-Although the procedure and images are well described, the tortuosity of the descending aorta in the present case did not look so bad shape. There was one acute kinking in the abdominal aorta but less calcification. For example, Figure 2C shows the straightened descending aorta, in which they had only one stiff wire and a pigtail catheter.

-Please consider showing the distribution of calcification.

-Please discuss the choice for THV prosthesis. Do the authors think which prosthesis would be ideal in a case with severe kinking? 

-How often does aortic dissection or rupture occur during TAVR, especially in such cases with severe kinking? Need for aortography or CT-angiography after the procedure? Please discuss.

Author Response

Dear Reviewers,

Thank you very much for your prompt reply. I very much appreciated and I find your questions precise and appropriate. I hope I have been exhaustive in responding to all comments made by the reviewers.

Please get back to me with any further questions you may have.

Best regards,

Alfredo Intorcia

1.Although the procedure and images are well described, the tortuosity of the descending aorta in the present case did not look so bad shape. There was one acute kinking in the abdominal aorta but less calcification. For example, Figure 2C shows the straightened descending aorta, in which they had only one stiff wire and a pigtail catheter.

The extreme tortuosity of the Descending Aorta are clearly visible in CT-Images both in a coronar view, with an acute angle of 75°, and in the sagittal view, with an acute angle calculated of 95°. The Back-up Meier "support" guidewire was held in place until the delivery system was able to pass through the tortuosity points of the Abdominal Aorta and then quickly removed to avoid some complications such as Aortic dissections or perforations.

2.Please consider showing the distribution of calcification.

Some calcified atherosclerotic plaque are clearly visible in the figure 1B e 1D in the CT 3D-volume rendering view all along the Descending Aorta and both the Iliac and femoral artery.

3.Please discuss the choice for THV prosthesis. Do the authors think which prosthesis would be ideal in a case with severe kinking?

For this purpose we choosed the Sapien3 THV prosthesis and its Ultra-Low profile 14 Fr Edwards eSheath (Edwards LifeSciences, Irvine, CA) introducer system; this sheat expands transiently during the passage of the Sapien 3 THV faciliting the placement of the large femoral sheat into the  atheroma-altered arterial-wall and tortuous iliaco-femoral and Aorta vessels of the patient.

4.How often does aortic dissection or rupture occur during TAVR, especially in such cases with severe kinking? Need for aortography or CT-angiography after the procedure? Please discuss.
Fortunately, in literature the reported incidence of acute Aortic dissection during TAVR is a rare complications involving approximately 0.2% of the cases
[15]. We routinely perform an ascending Aorta aortography at the end of all the procedure to avoid the above mentioned complications.

Round 2

Reviewer 2 Report

Congratulations.

This manuscript is a resubmission of an earlier submission. The following is a list of the peer review reports and author responses from that submission.